# The Unheeded Layers of Health Inequity: Visible Minority and Intersectionality

**DOI:** 10.3390/ijerph22071007

**Published:** 2025-06-26

**Authors:** Nashit Chowdhury, Tanvir C. Turin

**Affiliations:** 1Department of Community Health Sciences, Cumming School of Medicine, University of Calgary, Calgary, AB T2N 4N1, Canada; nashit.chowdhury@ucalgary.ca; 2Department of Family Medicine, Cumming School of Medicine, University of Calgary, Calgary, AB T2N 4N1, Canada

**Keywords:** health disparities, visible minority, intersectionality, structural inequities, systemic racism, social determinants of health

## Abstract

Health disparities among marginalized populations persist in many developed countries despite substantial population health advancements, highlighting persistent systemic inequities. Visible minorities, defined as the non-White and non-Indigenous racialized population in Canada, face earlier disease onset, worse outcomes, barriers to care, and shorter life expectancy. Conventional single-axis research frameworks, which examine factors like race, gender, or socioeconomic status in isolation, often fail to capture the complex realities of these disparities. Intersectionality theory, rooted in Black feminist thought and Critical Race Theory, offers a crucial lens for understanding how multiple systems of oppression intersect to shape health outcomes. However, its application in health research remains inconsistent, with often inadequate and tokenistic applications of this theory attributable to the limitations of a research approaches and resources, as well as biases from researchers. Integrating intersectionality with other relevant frameworks and theories in population health, such as ecosocial theory that explains how social inequalities become biologically embodied to create health inequities, strengthens the capacity to analyze health inequities comprehensively. This article advocates for thoughtful application of intersectionality in research to understand health disparities among visible minorities, urging methodological rigor, contextual awareness, and a focus on actionable interventions. By critically embedding intersectional principles into study design, researchers can move beyond describing disparities to identifying meaningful, equity-driven solutions. This approach supports a deeper, more accurate understanding of health inequities and fosters pathways toward transformative change in public health systems.

## 1. Introduction

Health disparities persist among marginalized populations despite many advancements made in population and public health. This indicates deeply entrenched systemic inequities within contemporary societies including western developed countries such as Canada [1,2]. These disparities manifest as early onset, worse prognosis, and higher prevalence of chronic diseases, barriers to accessing care, and shorter life expectancies among marginalized groups compared to their counterparts [2,3]. “Visible minorities” is a term particularly used in Canada to describe non-White, non-Indigenous racialized groups, typically including individuals of Arab, Asian, Black, Hispanic/Latinx, South Asian, and other diverse ethnic backgrounds, regardless of whether they were born in or immigrated to Canada [4,5]. The health inequities in western developed countries are particularly pronounced in these groups which are not isolated phenomena but emerge from complex interactions between multiple social determinants of health including systemic racism, labor market exclusion, language and cultural barriers, and immigration related stressors [3,6]. These determinants differ both in nature and origin from those affecting Indigenous peoples—whose health outcomes are largely shaped by settler colonialism, intergenerational trauma, and land dispossession—and from White populations, who are less likely to encounter racial discrimination or structural exclusion [7].

Conventional research approaches to understand health disparities have often employed single-axis frameworks, examining the influence of race, gender, or socioeconomic status on health in isolation [8,9,10]. While this approach informs us how these determinants individually shape health outcomes, the understanding of simultaneous effects of multiple determinants on people has been limited. Interventions based on such research outcomes may result in the so called “inequality paradox,” population-wide health interventions often inadvertently widen the gap by disproportionately benefiting privileged groups while leaving vulnerable populations behind [11].

Intersectionality theory by Crenshaw has emerged as a critical framework for addressing these limitations [12]. Originally inspired and built upon Black feminist thought and Critical Race Theory (CRT), intersectionality points out how multiple interlocking systems of oppression act collectively to construct unique experiences of marginalization [13,14]. This framework is critical to examine prevailing health disparities among visible minorities, as it informs the researchers of complex ways in which various social positions intersect to influence health outcomes.

The need for intersectionality-informed research design becomes particularly acute considering the lack of optimal healthcare access and worse health outcomes for visible minority communities [1]. However, the application of intersectionality theory in health research remains quite open and largely up to the researchers that act as a source of both strength and weakness depending on the approach of individual researchers. Harper et al. assert that many quantitative measurements of health inequities involve implicit value judgments by researchers in selecting variables, defining categories, and choosing statistical methods, which can bias the findings if the researchers are not cognizant of intersectionality [15]. Similarly, in qualitative studies, researchers’ view of inequity may lead to misinterpretation of the disparities experienced by the population [16].

Further, it is important to acknowledge that while intersectionality offers as a powerful tool to examine health disparities among visible minorities, the researchers need to resort to other health, social, and behavioral theories and frameworks that bring multiple perspectives and insights to explain observed disparities and inform an appropriate methodological approach. McLaren and Hawe emphasize the importance of ecological perspectives in health research, highlighting how human body and social and physical environment must be understood through their interdependencies and adaptations with the changes [17]. This framework aligns with intersectionality’s emphasis on examining how multiple social categories and systems of power interact within broader societal contexts to influence health outcomes. This necessitates research approaches that can capture the multidimensional nature of health inequities while identifying actionable intervention points.

Careful consideration and use of intersectional elements in research design can be crucial to better understand and address the complex health challenges faced by visible minority populations [3,18,19]. With this aim, this article presents critical considerations for designing research that applies intersectionality theory to study health disparities among visible minority populations. Specifically, we explore the following:The theoretical foundations and evolution of intersectionality theory as applied to health and wellness research;Methodological considerations for operationalizing intersectionality in both qualitative and quantitative research designs;The strengths and limitations of applying intersectionality as a theoretical framework in research, including potential complementary theoretical orientations.

## 2. Comprehensive Overview of Intersectionality Theory

### 2.1. Historical Development and Core Concepts

Intersectionality theory emerged from the recognition that traditional social frameworks failed to capture the complexity of marginalized experiences, particularly those of Black women. The theory originated from Black feminist thought and CRT, with Kimberlé Crenshaw coining the term to describe how the legal system’s treatment of race and gender discrimination as separate categories made Black women’s experiences invisible [12]. Crenshaw pointed out how traditional anti-discrimination frameworks, by treating race and gender as mutually exclusive categories in legal scenarios, systematically failed to account for the compounded forms of discrimination faced by the marginalized groups [12]. For example, implementing racial and gender equality in hiring process in isolation, Black women often remain disadvantaged as White women may still benefit from racial privilege and Black men from gender privilege. Williams et al. conceptualize these systems as organized social hierarchies where dominant groups maintain power through both ideological and structural means by allocating resources unequally, creating self-perpetuating cycles of disadvantage operating at multiple levels [5].

The evolution of intersectionality from legal theory to a broader social analysis framework represents a significant theoretical advancement in understanding complex social inequities. Following Crenshaw’s foundational work, intersectionality theory evolved through multiple disciplinary pathways before reaching public health. Scholars like Patricia Hill Collins and Deborah King’s earlier works on “multiple jeopardy” and “matrix of domination” further bolstered the foundation of intersectionality theory, while ethnic studies scholars incorporated analyses of citizenship, immigration status, and cultural identity [20,21]. Medical sociologists in the 1990s began applying intersectional frameworks to health disparities, examining how race, class, and gender jointly shaped health outcomes [19]. These disciplinary developments provided crucial methodological innovations—later formalized into approaches such as intracategorical and anticategorical frameworks—that would eventually inform intersectional approaches in public health research [22]. The literature on this theory soon expanded beyond Black women to other racial and other social categories. For instance, Latino Critical Theory (LatCrit) emerged in the mid-1990s, rooted in Chicana feminist thought, to address how axes of nativity, language, immigration status, and phenotype intersect with race and ethnicity to create unique forms of marginalization [23,24]. LatCrit’s emphasis on multidimensional identities and borderlands consciousness [25,26] expanded intersectionality’s analytical scope beyond the Black–White dichotomy, providing essential insights for understanding health disparities among visible minorities in contexts of migration and transnationalism.

Social categories beyond race and gender were started to be considered, such as sexuality, disability, migration status, etc. [27,28]. Further, the utility of this theory in understanding disparities in social work, health, and other areas were unpacked. Mykhalovskiy et al. described that population health in particular benefited from intersectionality by shifting focus from individual behaviors to structural determinants [18]. A systematic review by Harari and Lee demonstrates how health inequities arise not just from individual factors but from the complex interplay of various social positions and structural forces [3]. For example, a study by Pérez et al., included in that review, found higher odds of poor mental health for sexual minority women compared to heterosexual women [29]. This example underscores that intersectionality challenges population health researchers to think beyond simple categorizations of social variables to understand how multiple systems of power and oppression interact to create health disparities among marginalized communities such as visible minorities.

### 2.2. Theoretical Foundations for Intersectionality in Health and Wellness Research

Patricia Hill Collins’ concept of the “matrix of domination” deepens understanding of intersectionality by exploring how oppression functions simultaneously at structural, disciplinary, cultural, and interpersonal levels; specifically, this concept illuminates how these domains of power interweave to create interlocking systems of social control that cannot be dismantled through single-axis interventions [14,27]. By analyzing race, gender, and class as mutually constitutive, this concept shifts the focus to how domination affects individuals in multiple and interlocking forms. Similarly, CRT emphasizes the structural and systemic nature of racism, arguing that racism is deeply embedded within the legal, political, economic, and health systems [13]. Racism is both overtly and covertly active in our everyday practices and counter-narratives from marginalized communities are essential to challenge dominant ideologies. In population health, while Geoffrey Rose’s theory advocates us about addressing the structural determinants and taking a population level approach to address health inequities, intersectionality points out how different parts of the population would have different needs, strengths, limitations, and need for change in the system accordingly [30].

Williams et al. describes that health equity researchers build upon intersectionality framework to examine how structural racism operates as a system of power that not only categorizes individuals but also allocates health resources unequally [5]. McLaren and Hawe underscore how ecological perspectives in health research must consider how power dynamics operate across multiple levels, from individual interactions to broader societal structures [17]. Structural, political, and representational intersectionality, were further contextualized and elaborated by Mykhalovskiy et al.’s analysis, offering distinct but complementary lenses for understanding health inequities [18].

Structural intersectionality examines how social institutions and policies create overlapping disadvantages. As Homan et al. demonstrate in their work, this involves analyzing how multiple systems of stratification—such as racial residential segregation intersecting with gendered labor markets—produce compounded health disadvantages that cannot be understood through single-axis frameworks [31]. Their analysis reveals how structural intersectionality offers new directions for health disparities research by focusing on interlocking institutional arrangements rather than individual characteristics. Political intersectionality refers to how political movements and agendas often marginalize groups oppressed by multiple systems by prioritizing single-issue frameworks, such as feminist movements centering white women’s experiences or anti-racist movements centering men of color [32]. In health policy, this manifests when interventions designed for ‘women’ or ‘racial minorities’ fail to address the specific needs of women of color [33]. Representational intersectionality addresses how cultural narratives, media portrayals, and stereotypes shape public understanding and policy responses to health issues affecting multiply the effects for marginalized groups. This includes how medical research represents (or erases) certain populations and their health needs [34]

Power relations and social categories form central components of intersectional analysis [11]. Following Friel et al., we understand power as the capacity of actors (whether individuals, groups, or institutions) to influence [35], direct, or control the behavior of others and the course of events [36]. Harris et al. further distinguish between power over (domination), power to (agency), power with (collective action), and power within (empowerment) [37]. In health contexts, power operates through ideational mechanisms—shaping what is considered possible or legitimate—as well as material mechanisms that control resource distribution [36]. As Heller et al. argue, public health must recognize, analyze, and shift power relations to advance health and racial equity, moving beyond documenting disparities to transforming the systems that create them [38].

## 3. Research Design Considerations

### 3.1. Methodological Framework Considerations

The operationalization of intersectionality within health research warrants methodological frameworks that can adequately tackle the complex phenomenon of social inequities from epistemological and ontological orientation to data collection to interpretation of the research [8,39,40]. Researchers argued how intersectionality-informed study design must transcend traditional methodological boundaries, advocating for approaches that illuminate both individual lived experiences and structural determinants of health disparities [3]. Mykhalovskiy et al. emphasizes how research design must deliberately resist the hidden trenches created by the colonial system and consciously avoid their research being used to compound existing patterns of inequities [18]. This epistemological stance requires researchers to critically examine their own assumptions while designing their research that is equipped enough to capture the complexity of intersectional experiences [16,41,42]. Particularly, in research concerning visible minority populations the researchers need to be aware of the power relations and combat the risk of epistemic erasure—the potential for research methodologies to inadvertently silence or misrepresent marginalized voices [18].

Community engagement emerges as a critical foundation for intersectional approach in research design. It requires understanding community ecosystems and addressing both individual experiences and broader sociocultural contexts [43]. This approach aligns with what Whitehead identifies as typology of actions including targeting individual empowerment, community strengthening, improved living conditions, and macro-level policy changes can address both immediate health needs and underlying structural inequities [44]. Further, decisions about data collection, outcome measures, and interpretation can perpetuate or challenge inequities. For example, standardized questionnaires may not capture cultural nuances associated with a health risk factor (e.g., smoking) under investigation, thereby reinforcing victim-blaming [15]. Likewise, researchers need to be cognizant in choosing and applying statistical methods in their analysis. Scott and Siltanen demonstrate that integrating a feminist perspective into regression analysis acknowledges power dynamics, incorporates qualitative data, challenges objectivity, prioritizes community engagement, and focuses on actionable insights to promote social justice [39]. Table 1 provides some key insights for intersectional research design considerations covered in this section.

### 3.2. Qualitative Design Considerations

Qualitative methodologies by their nature offer a rich platform for capturing nuanced intersectional experiences. Certain approaches in sampling of participants can be useful to integrate intersectional mindsets in qualitative research. Harari and Lee advocate for purposive sampling strategies within qualitative research to ensure representation across diverse intersectional positions, including those based on race, gender, class, sexual orientation, and disability [3]. However, researchers must exercise careful attention to cultural safety and power dynamics throughout the research process. Scholars emphasize the importance of actively resisting the reproduction of existing power hierarchies and creating safe spaces for authentic expression [18,27,45]. Acknowledging and putting forward researchers’ positionality within broader systems of power and privilege also makes it transparent and supports creating space for voices traditionally marginalized by dominant research paradigms [42,46].

Effective communication and engagement require careful attention to cultural nuance. Researchers highlight that translation of research materials must transcend mere linguistic equivalence to capture nuanced cultural meanings and lived experiences [27,43,47]. Member checking, returning findings to participants for feedback and validation, further enhances rigor and ensures accurate representation of lived experiences, particularly crucial in intersectional research where power dynamics can influence interpretation. Intersectional training for the researchers and meaningful community engagement can enhance both the validity and relevance of research findings [43]. The training may include the following: (1) acquisition of foundational knowledge on intersectionality’s theoretical roots and power-analysis frameworks; (2) development of methodological skills for intersectional inquiry; and (3) cultivation of reflexive attitudes and commitments to co-production, community engagement, and anti-oppression principles [48,49]. By prioritizing community engagement, reflexivity, and cultural sensitivity, qualitative research can play a powerful role in amplifying marginalized voices and promoting health equity.

### 3.3. Quantitative Design Considerations

Quantifying intersectionality poses distinct methodological challenges that warrant uptake of innovative analytical approaches. Further, if the researchers are not informed of intersectionality when they design the study and make the analysis plan, their implicit value judgments may be embedded within seemingly objective measurement strategies, potentially obscuring the multidimensional nature of health disparities [15]. For example, treating “Asian” as a single category conceals critical heterogeneity: Chinese, Filipino, and South Asian subgroups often exhibit markedly different health profiles, yet these differences are lost when data are aggregated into broad racial labels [48]. The selection of reference groups in regression models constitutes another value-laden choice—defaulting to White populations as the normative referent not only centers whiteness in interpretation but also influences the relative magnitude of estimated effects for all other groups [33]. However, many studies involve working with secondary data where the scope of taking intersectional approaches are often limited due to initial design restrictions.

One key challenge lies in the frequent lack of self-reported demographic data for quantitative research. Elliott et al. attempts to address this challenge by their groundbreaking work on indirect estimation methods for frequently missing or inaccurate race/ethnicity data, introducing the Bayesian Surname and Geocoding approach [50]. Further methodological adaptations are necessary to address the limitations of conventional sample size and power calculations, particularly when dealing with smaller subgroups within intersectional research. Lederer et al. detail how traditional power calculations often fail to capture intersectional effects, especially when investigating multiple, overlapping social positions [51]. Their analysis advocates for methodological innovations that can accommodate smaller subgroup samples while maintaining statistical rigor. The choice of statistical models must align with intersectional theory, as demonstrated by Dill & Kohlman who propose interaction-centered modeling, multilevel frameworks, and the integration of contextual variables at multiple scales to capture complex social realities [14]. For example, using Multilevel Analysis of Individual Heterogeneity and Discriminatory Accuracy (MAIHDA) can help overcome reduced statistical power to detect effects and account for nested nature of intersectional social categories [8]. MAIHDA treats intersectional strata (such as combinations of race, gender, and class) as random effects, enabling researchers to quantify how much health variation exists between different intersectional groups versus within them, while fixed effects estimate overall population patterns. This approach allows researchers to separate additive effects of individual social categories from true intersectional effects that emerge from their unique combinations. Incorporating temporal dynamics into statistical models further enhances the ability to analyze how intersectional experiences unfold over time.

Mahendran et al. conducted a comprehensive simulation study comparing six methods for estimating health outcomes across high-dimensional intersections: cross-classification, regression with interaction terms, MAIHDA, classification and regression trees (CART), conditional inference trees (CTree), and random forest [52]. Their findings demonstrate that method choice significantly impacts outcome estimates—in their National Health and Nutrition Examination Survey (NHANES) example, different methods resulted in up to 10 mmHg difference in predicted systolic blood pressure for the same intersectional group, a clinically meaningful difference. For smaller sample sizes, MAIHDA, CTree, and random forest produced more accurate and stable estimates, while CART consistently performed poorly across all sample sizes. Importantly, this evaluation emphasizes that purely data-driven analysis must be balanced with theoretical knowledge about social power structures to ensure that identified intersections remain socially meaningful intervention points rather than merely statistical artifacts.

Finally, interpreting findings in intersectional research requires moving beyond statistical significance to substantive meaning. Scholars emphasize the importance of contextualizing results within broader social patterns, using triangulation (i.e., integrating insights from multiple methods and data sources), and acknowledging the limitations of traditional significance testing [53]. Recognizing the dynamic nature of intersectional effects is crucial for developing nuanced interpretations that accurately reflect the lived experiences of individuals and communities.

### 3.4. Strengths and Limitations of Applying Intersectionality as a Theoretical Framework

The strength of employing intersectionality theory to health disparities research lies in its ability to illustrate how complex interplay of social positions and systems of power influence health. For instance, the World Health Organization’s conceptual framework for action on social determinants of health provides and recognizes social positions including identities like race and gender as structural determinants of health, thereby shifting the focus from making changes within individuals to changing systems for better health outcomes [2]. Phelan et al. in their ‘Fundamental Cause Theory’ points out that while the social positions may determine health, the fundamental causes of health disparities are the discriminatory systems based on these positions such as racism, sexism, classism, and other systems of oppression [54]. These systems operate as ‘causes of causes’ by structuring access to flexible resources like money, knowledge, power, prestige, and beneficial social connections. The theory explains how health disparities persist because these oppressive systems continuously shape the distribution of resources that can be deployed to avoid risks or minimize consequences of disease. For instance, racism as a fundamental cause explains that even as specific mechanisms of racial health disparities are addressed (e.g., neighborhood segregation), new mechanisms emerge (e.g., algorithmic bias in healthcare) to maintain the disadvantage. Intersectionality informs us that even if the effects of one social position, such as income, are minimized the resource allocation linked to other social positions, such as race and education, perpetuate the same disparities in health.

Intersectionality does not necessarily demand applying a specific and narrowly defined framework in research. This acts as a double-edged sword by offering flexibility while simultaneously risking superficiality. This flexibility allows researchers to tailor their approach to the specific context and community under study, fostering innovation and responsiveness. However, this opens up an avenue for superficial applications of intersectionality, where researchers may simply report how certain outcomes are correlated with multiple demographic variables without thoughtful illustration of how the complex power dynamics and social inequities interact to result their findings [14]. Without a strong theoretical grounding and a commitment to rigorous analysis, intersectionality can become a buzzword rather than a transformative analytical tool and eventually reinforce existing inequities.

Balancing theoretical sophistication with practical applicability is crucial but difficult. Studies emphasized how the selection of intersectional frameworks must align with specific research objectives while remaining attentive to power dynamics within research processes. For example, while MAIHDA offers an excellent statistical method to incorporate intersectionality in quantitative research [8], the challenge lies in ensuring that mathematical operationalization of intersectionality captures the nuanced, non-additive dynamics central to intersecting systems of oppression while remaining methodologically rigorous and interpretively accessible. This warrants that framework selection process carefully balance methodological sophistication with interpretative accessibility.

Resource implications also pose a significant challenge [54]. Meaningful community engagement and partnership building with the community require significant investment in time, resources, and relationship building [43]. Similarly, capacity building within both research teams and community partner organizations is essential for ensuring that research is conducted ethically and sustainably. These resource demands can create barriers for researchers, particularly those working with limited funding or within resource-constrained settings.

## 4. Theoretical Integration Possibilities

While conducting a particular research project or running a program of research, the integration of complementary theoretical frameworks to intersectionality can enhance our understanding of contemporary health disparities among visible minority populations. CRT, which shares foundational synergies with intersectionality can further critically complement the investigation of health disparities, particularly for visible minorities [13]. While intersectionality informs about existing multiple systems of oppression, having the CRT lens allows researchers to trace how racial constructs intersect with other axes of oppression to produce distinctive health vulnerabilities, offering crucial insights for both research design and intervention strategies. For example, Small et al. demonstrated how combining intersectionality with CRT’s concept of ‘racial realism’ (i.e., racism being inherently ordinary, not the exception, and embedded into systems) revealed how Black and Latina women’s experiences in sexual healthcare settings were rendered invisible through ‘biological individualism’—where providers focused solely on individual risk behaviors rather than recognizing how systemic racism and gendered oppression shaped health vulnerabilities [55],. This dual framework exposed both the intersectional nature of their marginalization and the specific mechanisms through which institutional racism operates to perpetuate health disparities.

Feminist theory, which fundamentally examines how gender functions as a system of power relations and social organization, significantly enriches intersectional health disparities research. Scott and Siltanen in their analysis reveal how feminist theoretical perspectives can enhance intersectional frameworks by focusing on how gender operates as a fundamental organizing principle in health systems [39]. For example, their findings revealed that South Asian women in Toronto showed significantly higher odds of reporting poor health compared to white women and men, while this pattern did not emerge in New York City. The feminist theoretical framework proved especially valuable in uncovering how individual-level gender effects intersect with neighborhood contexts to influence health disparities.

Decolonial theoretical perspectives offer essential insights that can strengthen intersectional analysis. Thambinathan and Kinsella’s framework reveals how traditional research paradigms often perpetuate epistemic erasure while simultaneously affecting material including health outcomes for marginalized communities [56]. Integrating decolonial theory with intersectionality equips the researchers to address the limitations of non-Western ways of knowing and complement the research with Indigenous and other ways of knowing that are more sensitive to capture the intersectional effects of health on visible minorities. This theoretical integration helps researchers understand how colonial legacies continue to systematically marginalize the visible minority population’s lived experiences and ways of knowing, and shape health disparities while identifying pathways for decolonial health interventions. When merged with intersectionality, decolonial perspectives advocate for what Thambinathan and Kinsella term as “transformative praxis,” moving beyond theoretical analysis to demand concrete systemic change [56]. This theoretical synthesis illuminates pathways for resistance while maintaining cultural integrity, particularly salient for visible minority communities navigating multiple systems of oppression. Through this integrated lens, researchers can better understand and address the complex interplay of historical trauma, contemporary discrimination, and systemic barriers, while identifying opportunities for transformation and healing.

Young’s framework of the ‘five faces of oppression’ offers another valuable theoretical integration with intersectionality for health disparities research [57]. Young’s taxonomy—exploitation, marginalization, powerlessness, cultural imperialism, and violence—provides a structural analysis of the mechanisms through which social inequities are produced and maintained [58]. When integrated with intersectionality, researchers can identify both who bears compounded burdens of inequity (via intersectionality) and the specific forms that oppression assumes in their lived contexts (via Young’s framework). For example, Ayón et al. found how the intersecting identities of immigrant Mexican mothers in Arizona created unique vulnerabilities across all five faces of oppression manifested through wage disparities, exclusion from health services, feeling unable to report harassment due to documentation status, shaming for speaking Spanish and traditional childrearing practices, workplace sexual harassment, and more [59].

Biosocial and ecosocial theoretical approaches can also complement intersectional analysis [60]. Specifically, Krieger’s ecosocial theory provides a framework for understanding the biological pathways through which societal inequalities produce health disparities, asking who and what is responsible for these patterns. McLaren et al. shows how population strategies for health improvement must consider different social positions interact to create unique health challenges [30]. The concept of embodiment plays a crucial role in intersectional health research. As social conditions become literally embodied in biological outcomes, with discrimination and structural inequities “getting under the skin” to produce health disparities. For instance, the chronic stress from experiencing both racial and gendered discrimination is not just a psychological state; it can be embodied over the course of life as chronic inflammation or hypertension. This biological embedding of social disadvantage helps explain how intersecting forms of oppression contribute to differential health outcomes.

## 5. Conclusions

The overview and application of intersectionality theory to examine health disparities research among visible minority populations generates crucial insights while illustrating pathways for future investigation. Both quantitative and qualitative methodological frameworks must consciously balance theoretical sophistication with practical applicability. Population health research approaches need to adapt to avoid inadvertently exacerbating existing inequities. Critical success factors for intersectional health research include meaningful community engagement, methodological innovation, and careful attention to power dynamics. Racism and sexism functions as a structured system of power. To untangle these systems of oppression, methodological innovations need to be continued. As health disparities research continues to evolve, intersectional frameworks will undoubtedly play an increasingly crucial role in advancing our understanding of how multiple systems of advantage and disadvantage interact to shape health outcomes. This understanding, in turn, can inform more effective and equitable health interventions for visible minority populations and all marginalized communities.

## 6. Authors’ Positionality

As researchers situated within Canadian academic institutions, we acknowledge our social locations shape the perspective laid out in this article. The first author (NC) is a racialized immigrant researcher whose lived experiences of navigating multiple marginalized identities inform the theoretical engagement with intersectionality. The second author (TCT) is also a racialized immigrant researcher and brings expertise in population health and health equity research among immigrant and racialized populations. Both authors work within Western academic frameworks while seeking to challenge their limitations. We recognize that our institutional positions afford certain privileges that may influence our perspectives, and we strive to remain reflexive about these throughout our analysis.

## Figures and Tables

**Table 1 ijerph-22-01007-t001:** Key considerations for intersectional research design.

Research Phase	Qualitative Approaches	Quantitative Approaches
Conceptualization	• Engage community partners early • Map power relations in research context • Consider historical/political context • Reflexivity about researcher position	• Move beyond demographic controls • Plan for interaction effects • Consider sample size for subgroups
Data Collection	• Purposive sampling across intersections • Develop cultural safety protocols • Create spaces for safe expression • Culturally validated instruments	• Collect disaggregated data • Include contextual variables • Consider proxy measures • Oversample marginalized groups
Analysis	• Develop intersectional coding schemes • Analyze power in narratives • Look for silences/absences • Use member checking	• Test interactions systematically • Apply analytical method that accommodates intersections • Consider alternative analytical method beyond conventions
Interpretation	• Center marginalized voices • Identify structural barriers • Link to broader systems • Avoid damage-centered narratives	• Consider practical relevance • Contextualize within power structures • Ensure that the interpretation does not further marginalize

## Data Availability

No new data were created or analyzed in this study. Data sharing is not applicable to this article.

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
