# Peer review of "The Unheeded Layers of Health Inequity: Visible Minority and Intersectionality"

_ijerph, 2025, doi:10.3390/ijerph22071007_

Round 1
Reviewer 1 Report
Comments and Suggestions for Authors
This perspective style article provides a high-level overview of intersectional theory. Specifically, this article seeks to accomplish three aims: (1) summarize the foundations of and historical evolution of intersectional theory, (2) provide a brief overview of methodological consideration needed for conducting intersectional qualitative and quantitative work, and (3) address the strengths, limitations, and complementary theoretical frameworks. I commend the authors for this undertaking. I fully agree on the importance of intersectional theory and the need for conceptually and methodologically rigorous intersectional work to achieve health equity globally. However, given the intention of the article, there are several areas that could use additional theoretical grounding and elaboration, as well as reorganization of content among the sections. Specific suggestions and evaluation of each are below.
General concept comment:
Per the title, abstract, and introduction, the orientation of this article is to “visible minorities,” implicitly defined as “non-White and non-Indigenous racialized groups.” I find the use of “visible minorities” to be overly vague and the exclusion of Indigenous peoples from being a “visible minority” curious. As a note of positionality: my knowledge of racial formation and systemic oppression is rooted in US and Latin American contexts. Therein, Indigenous peoples are very frequently cast as a “visually identifiable ‘other’.” On being overly broad, “visible minorities” seemingly could also include those with physical disabilities (which based on my reading of the work is not the intended focus).
I recognize that perhaps this terminology (“visible minorities”) is specific to the Canadian context. However, as this is a journal with an international readership, I recommend either (1) a more specific definition and explanation as to why the term “visible minorities” is being used early in the introduction or (2) substitution for a different phrasing (e.g., “minoritized racial groups,” “minority racialized groups,” etc.).
Specific Comments:
- [Line 96- 127] Section “1.2 Historical Developments and Core Concepts” jumps from 1989 to work in 2014-2021. I appreciate the need for brevity, however, as part of the “comprehensive overview of intersectional theory” this feels incongruous.
- For example, from the legal space, intersectional theory made its way into the disciplinary practices of sociology (particularly medical sociology) and ethnic studies much earlier than public health. Some of the thinking about how to apply intersectionality both quantitatively and qualitatively stems from work done by scholars in these other disciplines. Given the inherently interdisciplinary history, I believe at least brief recognition of how intersectional theory has been further developed in other disciplines is needed.
- Furthermore, this summary entirely missed the development of Latino Critical Theory, an intersectional extension of CRT specifically rooted in Chicana Feminist thought to address the axes of nativity, immigration, culture. Again, I belief some (albeit brief) mention of Latino Critical Theory needed in this discourse.
- [Line 128-210] In section “1.3. Theoretical Foundations for Intersectionality in Health Research” I was expecting a further summary and elaboration into how feminist thought, CRT, and theorization of power gave rise to intersectional theory and its application in public health research. This is somewhat accomplished; however, this is where I believe the theoretical grounding of the work could be most improved.
- Specifically in relation to the discussion of power: While being a central concept to the article and referenced throughout the work – power is not defined at all. Some references for further conceptualizing power:
- Friel S, Townsend B, Fisher M, et al. Power and the people’s health. Social Science & Medicine. 2021;282:114173.
- Harris P, Baum F, Friel S, et al. A glossary of theories for understanding power and policy for health equity. J Epidemiol Community Health . 2020;74(6):548–552.
- Heller JC, Fleming PJ, Petteway RJ, et al. Power Up: A Call for Public Health to Recognize, Analyze, and Shift the Balance in Power Relations to Advance Health and Racial Equity. Am J Public Health. 2023;113(10):1079–1082.
- Carstensen MB, Schmidt VA. Power through, over and in ideas: conceptualizing ideational power in discursive institutionalism. Journal of European Public Policy. 2016;23(3):318–337.
- [lines 150-156] I recommend expanding the discussion of structural, political and representational intersectionality and incorporating additional perspectives beyond Mykhaloviskiy et al (2019) paper.
- For example, see: (Homan P, Brown TH, King B. Structural Intersectionality as a New Direction for Health Disparities Research. J Health Soc Behav. 2021;0(00):1-21. doi:10.1177/00221465211032947) on structrual intersectionality.
- Also please define political intersectionality.
- Specifically in relation to the discussion of power: While being a central concept to the article and referenced throughout the work – power is not defined at all. Some references for further conceptualizing power:
- [line 158] The paragraph beginning 158 mentions power and then leads into an example. However, the discussion of the example does not actually explicitly lead back to an analysis of power. Please make the connection between power relations and this example more explicit.
- [lines 171-177] I recommend moving the discussion of biosocial approaches to section 3.5 on “Theoretical Integration possibilities,” as it is not a foundational component of intersectional theory. This may also be extended to the referencing of ecosocial perspectives [line 147].
- Per the discussion on line 189-192 and lines 221-224, I recommend that the authors provide their own positionality statements. Generally, per the authors’ own points, positionality statements are important for contextualizing the framing and methodological decision making of research. I believe this is especially so in international journals and would encourage IJERPH to make them a requirement. I defer to the editor on the best location for such statements (e.g., in an acknowledgements section outside of the main text, at the beginning or end of the main text).
- [Line 231] Please define or provide examples of “intersectional training.”
- [Line 281 – 289] Please revisit the summary of fundamental cause theory (FTC), specifically the sentence beginning on line 281. FTC does not attribute the “the cause of causes” to the social position itself but the social-structural mechanism which gives the position meaning – i.e., racism, classism, sexism.
- [Line 318] In order to better mirror the three aims, I recommend Section “3.5 Theoretical Integration Possibilities” be relabeled as section 4 and bolded. This creates a one-to-one match between the aims and main sections of the articles.
- [Line 319-341] Per my 2nd comment and the authors’ argument, CRT and feminist theory are part of the foundation of intersectional theory. I believe the content of these two paragraphs belongs in section 1.3 [beginning line 128] on the foundations of intersectional theory rather than in the section on parallel theories. For example, can an intersectional approach be applied without CRT and feminist theory, given they are the theoretical foundation of intersectionality itself?
- With respect to aim 3 as stated on line 93-94, the strengths and limitations of intersectional theory were not explicitly (or at all) addressed. I recommend a revision of the aims or a reorganization of the article to make this more direct.
Author Response
- This perspective style article provides a high-level overview of intersectional theory. Specifically, this article seeks to accomplish three aims: (1) summarize the foundations of and historical evolution of intersectional theory, (2) provide a brief overview of methodological consideration needed for conducting intersectional qualitative and quantitative work, and (3) address the strengths, limitations, and complementary theoretical frameworks. I commend the authors for this undertaking. I fully agree on the importance of intersectional theory and the need for conceptually and methodologically rigorous intersectional work to achieve health equity globally. However, given the intention of the article, there are several areas that could use additional theoretical grounding and elaboration, as well as reorganization of content among the sections. Specific suggestions and evaluation of each are below.
Reply:
We sincerely thank you for your positive assessment of our work and for recognizing the importance of intersectional theory in achieving health equity. We have carefully considered all your suggestions and have made substantial revisions to address each point, as detailed in our responses below.
- Per the title, abstract, and introduction, the orientation of this article is to “visible minorities,” implicitly defined as “non-White and non-Indigenous racialized groups.” I find the use of “visible minorities” to be overly vague and the exclusion of Indigenous peoples from being a “visible minority” curious. As a note of positionality: my knowledge of racial formation and systemic oppression is rooted in US and Latin American contexts. Therein, Indigenous peoples are very frequently cast as a “visually identifiable ‘other’.” On being overly broad, “visible minorities” seemingly could also include those with physical disabilities (which based on my reading of the work is not the intended focus).
I recognize that perhaps this terminology (“visible minorities”) is specific to the Canadian context. However, as this is a journal with an international readership, I recommend either (1) a more specific definition and explanation as to why the term “visible minorities” is being used early in the introduction or (2) substitution for a different phrasing (e.g., “minoritized racial groups,” “minority racialized groups,” etc.).
Reply:
Thank you for highlighting this important conceptual issue and for sharing your positionality, which helps us understand the context of your critique. We recognize that the term "visible minorities" has specific legal and policy origins in the Canadian context that may not translate well internationally.
Yes, as you recognize Indigenous Peoples in Canada or people living with physical disabilities do not fall under the definition of the term 'visible minority' by the Employment Equity Act in Canada, which reflects their distinct constitutional status as the original inhabitants with inherent rights, treaty relationships, and nation-to-nation engagement with the Canadian state. This separation recognizes Indigenous sovereignty and self-determination while establishing different policy frameworks for addressing systemic discrimination.
We have revised the introduction to provide clearer definition and context.
Page 1, lines 39-52:
“Visible minorities” is a term particularly used in Canada to describe non-White, non-Indigenous racialized groups, typically including individuals of Arab, Asian, Black, Hispanic, South Asian, and other diverse ethnic backgrounds, regardless of whether they were born in Canada or immigrated (Employment Equity Act, 1995; Williams et al., 2019). The health inequities in western developed countries are particularly pronounced in these groups which are not isolated phenomena but emerge from complex interactions between multiple social determinants of health including systemic racism, labor market exclusion, language and cultural barriers, and immigration related stressors (Fehrenbacher & Patel, 2020; Harari & Lee, 2021). These determinants differ both in nature and origin from those affecting Indigenous peoples—whose health outcomes are largely shaped by settler colonialism, intergenerational trauma, and land dispossession—and from White populations, who are less likely to encounter racial discrimination or structural exclusion (Parter et al., 2023).
- [Line 96- 127] Section “1.2 Historical Developments and Core Concepts” jumps from 1989 to work in 2014-2021. I appreciate the need for brevity, however, as part of the “comprehensive overview of intersectional theory” this feels incongruous.
For example, from the legal space, intersectional theory made its way into the disciplinary practices of sociology (particularly medical sociology) and ethnic studies much earlier than public health. Some of the thinking about how to apply intersectionality both quantitatively and qualitatively stems from work done by scholars in these other disciplines. Given the inherently interdisciplinary history, I believe at least brief recognition of how intersectional theory has been further developed in other disciplines is needed.
Reply: Thank you for this important observation about the interdisciplinary evolution of intersectionality. We have added content addressing the trajectory through sociology and ethnic studies.
Page 3, lines 123-140:
Following Crenshaw's foundational work, intersectionality theory evolved through multiple disciplinary pathways before reaching public health. Medical sociologists in the 1990s began applying intersectional frameworks to health disparities, examining how race, class, and gender jointly shaped health outcomes (Weber & Parra-Medina, 2003). Scholars like Patricia Hill Collins (1990) and Deborah King (1988) expanded theoretical foundations through sociology, while ethnic studies scholars incorporated analyses of citizenship, immigration status, and cultural identity (Anzaldúa, 1987). These disciplinary developments provided crucial methodological innovations—such as intracategorical and anticategorical approaches (McCall, 2005)—that would later inform intersectional approaches in public health research.
- Furthermore, this summary entirely missed the development of Latino Critical Theory, an intersectional extension of CRT specifically rooted in Chicana Feminist thought to address the axes of nativity, immigration, culture. Again, I belief some (albeit brief) mention of Latino Critical Theory needed in this discourse.
Reply: You are absolutely correct that our overview missed this crucial development. We have added content on Latino Critical Theory.
Page 3, lines 132-140:
The literature on this theory soon expanded beyond Black women to other racial and other social categories. For instance, Latino Critical Theory (LatCrit) emerged in the mid-1990s, rooted in Chicana feminist thought, to address how axes of nativity, language, immigration status, and phenotype intersect with race and ethnicity to create unique forms of marginalization (Iglesias & Valdes, 2025; Shelton, 2018, p. 201). LatCrit's emphasis on multidimensional identities and borderlands consciousness (Fernández, 2002; Valdez & Lugg, 2010) expanded intersectionality's analytical scope beyond the Black-White dichotomy, providing essential insights for understanding visible minority health in contexts of migration and transnationalism.
- [Line 128-210] In section “1.3. Theoretical Foundations for Intersectionality in Health Research” I was expecting a further summary and elaboration into how feminist thought, CRT, and theorization of power gave rise to intersectional theory and its application in public health research. This is somewhat accomplished; however, this is where I believe the theoretical grounding of the work could be most improved. Specifically in relation to the discussion of power: While being a central concept to the article and referenced throughout the work – power is not defined at all. Some references for further conceptualizing power:
- Friel S, Townsend B, Fisher M, et al. Power and the people’s health. Social Science & Medicine. 2021;282:114173.
- Harris P, Baum F, Friel S, et al. A glossary of theories for understanding power and policy for health equity. J Epidemiol Community Health . 2020;74(6):548–552.
- Heller JC, Fleming PJ, Petteway RJ, et al. Power Up: A Call for Public Health to Recognize, Analyze, and Shift the Balance in Power Relations to Advance Health and Racial Equity. Am J Public Health. 2023;113(10):1079–1082.
- Carstensen MB, Schmidt VA. Power through, over and in ideas: conceptualizing ideational power in discursive institutionalism. Journal of European Public Policy. 2016;23(3):318–337.
Reply: Thank you for identifying this critical gap and providing these excellent references. We have added a comprehensive definition of power drawing from all the suggested sources.
Page 5, lines 198-208:
Power relations and social categories form central components of intersectional analysis (Frohlich & Potvin, 2008). Following Friel et al. (2021), we understand power as the capacity of actors (whether individuals, groups or institutions) to influence, direct or control the behavior of others and the course of events (Carstensen & Schmidt, 2016). Harris et al. (2020) further distinguish between power over (domination), power to (agency), power with (collective action), and power within (empowerment). In health contexts, power operates through ideational mechanisms—shaping what is considered possible or legitimate—as well as material mechanisms that control resource distribution (Carstensen & Schmidt, 2016). As Heller et al. (2023) argue, public health must recognize, analyze, and shift power relations to advance health and racial equity, moving beyond documenting disparities to transforming the systems that create them.
- [lines 150-156] I recommend expanding the discussion of structural, political and representational intersectionality and incorporating additional perspectives beyond Mykhaloviskiy et al (2019) paper.
For example, see: (Homan P, Brown TH, King B. Structural Intersectionality as a New Direction for Health Disparities Research. J Health Soc Behav. 2021;0(00):1-21. doi:10.1177/00221465211032947) on structrual intersectionality.
Also please define political intersectionality.
Reply: We have expanded this discussion and incorporated the suggested reference along with additional perspectives.
Page 4-5, lines 181-196:
Structural intersectionality examines how social institutions and policies create overlapping disadvantages. As Homan et al. (2021) demonstrate in their work, this involves analyzing how multiple systems of stratification—such as racial residential segregation intersecting with gendered labor markets—produce compounded health disadvantages that cannot be understood through single-axis frameworks. Their analysis reveals how structural intersectionality offers new directions for health disparities research by focusing on interlocking institutional arrangements rather than individual characteristics. Political intersectionality, as originally conceptualized by Crenshaw (2013), refers to how political movements and agendas often marginalize groups oppressed by multiple systems by prioritizing single-issue frameworks, such as feminist movements centering white women's experiences or anti-racist movements centering men of color. In health policy, this manifests when interventions designed for 'women' or 'racial minorities' fail to address the specific needs of women of color (Bowleg, 2012). Representational intersectionality addresses how cultural narratives, media portrayals, and stereotypes shape public understanding and policy responses to health issues affecting multiply marginalized groups. This includes how medical research represents (or erases) certain populations and their health needs (Schulz & Mullings, 2006)
- [line 158] The paragraph beginning 158 mentions power and then leads into an example. However, the discussion of the example does not actually explicitly lead back to an analysis of power. Please make the connection between power relations and this example more explicit.
Reply: Thank you for pointing out this mismatch. Given expansion and drawing from more extended literature based on your previous feedback, we removed this example.
- [lines 171-177] I recommend moving the discussion of biosocial approaches to section 3.5 on “Theoretical Integration possibilities,” as it is not a foundational component of intersectional theory. This may also be extended to the referencing of ecosocial perspectives [line 147].
Reply: We agree with this organizational suggestion and have relocated both biosocial and ecosocial discussions to the theoretical integration section.
Page 9, lines 438-445 (new location in Section 4):
Biosocial and ecosocial theoretical approaches can also complement intersectional analysis. McLaren et al. (2010) shows how population strategies for health improvement must consider different social positions interact to create unique health challenges. The concept of embodiment plays a crucial role in intersectional health research. As social conditions become literally embodied in biological outcomes, with discrimination and structural inequities "getting under the skin" to produce health disparities. This biological embedding of social disadvantage helps explain how intersecting forms of oppression contribute to differential health outcomes.
- Per the discussion on line 189-192 and lines 221-224, I recommend that the authors provide their own positionality statements. Generally, per the authors’ own points, positionality statements are important for contextualizing the framing and methodological decision making of research. I believe this is especially so in international journals and would encourage IJERPH to make them a requirement. I defer to the editor on the best location for such statements (e.g., in an acknowledgements section outside of the main text, at the beginning or end of the main text).
Reply: We completely agree with the importance of positionality statements and appreciate your advocacy for making them standard practice. We have added our positionality statement at the end of the manuscript.
Page 10-11, lines 464-475 (new section before Introduction):
Authors' Positionality: As researchers situated within Canadian academic institutions, we acknowledge our social locations shape the perspective laid out in this article. The first author (NC) is a racialized immigrant researcher whose lived experiences of navigating multiple marginalized identities inform the theoretical engagement with intersectionality. The second author (TCT) is also a racialized immigrant researcher and brings expertise in population health and health equity research among immigrant and racialized populations. Both authors work within Western academic frameworks while seeking to challenge their limitations. We recognize that our institutional positions afford certain privileges that may influence our perspectives, and we strive to remain reflexive about these throughout our analysis.
- [Line 231] Please define or provide examples of “intersectional training.”
Reply: We have added a comprehensive definition with specific examples as requested.
Page 6, lines 263-273:
Intersectional training for the researchers and meaningful community engagement can enhance both the validity and relevance of research findings (Wallerstein & Duran, 2006). The training may include: (1) acquisition of foundational knowledge on intersectionality’s theoretical roots and power-analysis frameworks; (2) development of methodological skills for qualitative, quantitative, and mixed-methods intersectional inquiry; and (3) cultivation of reflexive attitudes and commitments to co-production, community engagement, and anti-oppression principles (Sabik, 2021; Tinner et al., 2023). By prioritizing community engagement, reflexivity, and cultural sensitivity, qualitative research can play a powerful role in amplifying marginalized voices and promoting health equity.
- [Line 281 – 289] Please revisit the summary of fundamental cause theory (FTC), specifically the sentence beginning on line 281. FTC does not attribute the “the cause of causes” to the social position itself but the social-structural mechanism which gives the position meaning – i.e., racism, classism, sexism.
Reply: Thank you for pointing this out. We have revised our characterization of FCT to accurately reflect that systems of oppression, not social positions, are the fundamental causes.
Page 8, lines 339-352:
Phelan et al. (2010) in their 'Fundamental Cause Theory' points out that while the social positions may determine health, the fundamental causes of health disparities are the discriminatory systems based on these positions such as racism, sexism, classism, and other systems of oppression. These systems operate as 'causes of causes' by structuring access to flexible resources like money, knowledge, power, prestige, and beneficial social connections. The theory explains how health disparities persist because these oppressive systems continuously shape the distribution of resources that can be deployed to avoid risks or minimize consequences of disease. For instance, racism as a fundamental cause ensures that even as specific mechanisms of racial health disparities are addressed (e.g., neighborhood segregation), new mechanisms emerge (e.g., algorithmic bias in healthcare) to maintain the disadvantage. Intersectionality informs us that even if the effects of one social position such as income are minimized the resource allocation linked to other social positions such as race, education perpetuate the same disparities in health.
- [Line 318] In order to better mirror the three aims, I recommend Section “3.5 Theoretical Integration Possibilities” be relabeled as section 4 and bolded. This creates a one-to-one match between the aims and main sections of the articles.
Reply: We appreciate this organizational suggestion and have implemented it to create better alignment between our aims and structure.
- [Line 319-341] Per my 2ndcomment and the authors’ argument, CRT and feminist theory are part of the foundation of intersectional theory. I believe the content of these two paragraphs belongs in section 1.3 [beginning line 128] on the foundations of intersectional theory rather than in the section on parallel theories. For example, can an intersectional approach be applied without CRT and feminist theory, given they are the theoretical foundation of intersectionality itself?
Reply:
We appreciate your point about CRT and feminist theory being foundational to intersectionality. However, we respectfully maintain that this content belongs in Section 4 (Theoretical Integration Possibilities) as it aligns with our third objective to explore how intersectionality can be integrated with complementary theoretical frameworks in specific research contexts. While intersectionality indeed originated from CRT and feminist thought, we are discussing here how a particular research project or program of research focusing on certain health disparities can benefit from explicitly incorporating these theories in the contemporary context alongside intersectionality to deepen analysis. We have revised the text in Section 4 to clarify this distinction.
Page 9, lines 381-397:
While conducting a particular research project or running a program of research, the integration of complementary theoretical frameworks to intersectionality can enhance our understanding of contemporary health disparities among visible minority populations. CRT, which shares foundational synergies with intersectionality can further critically complement the investigation of health disparities, particularly for visible minorities (Delgado & Stefancic, 1998). While intersectionality informs about existing multiple systems of oppression, having CRT lens allows researchers to trace how racial constructs intersect with other axes of oppression to produce distinctive health vulnerabilities, offering crucial insights for both research design and intervention strategies. For example, Small et al. (Small et al., 2023) demonstrated how combining intersectionality with another tenet of CRT, racial realism (i.e., racism being inherently ordinary, not the exception, and embedded into systems) revealed how Black and Latina women's experiences in sexual healthcare settings were rendered invisible through 'biological individualism'—where providers focused solely on individual risk behaviors rather than recognizing how systemic racism and gendered oppression shaped health vulnerabilities. This dual framework exposed both the intersectional nature of their marginalization and the specific mechanisms through which institutional racism operates to perpetuate health disparities
- With respect to aim 3 as stated on line 93-94, the strengths and limitations of intersectional theory were not explicitly (or at all) addressed. I recommend a revision of the aims or a reorganization of the article to make this more direct.
Reply: Thank you for identifying this ambiguity. In fact, our third objective was to understand the strengths and limitations of applying intersectionality as a theoretical framework in research and it was covered in section 3.4 with a misleading heading. We now updated that heading, the third objective to clarify that.
New heading: 3.4. Strengths and Limitations of Applying Intersectionality as a Theoretical Framework
Reviewer 2 Report
Comments and Suggestions for Authors
I learned from reading this article, but most of what I learned came from the excellent explanation of the concept of intersectionality and how it applies to the analysis of health disparities. The article does not report the results of new research. It "advocates for thoughtful application of research on visible minority health, urging methodological rigor, contextual awareness, and a focus on actionable interventions." (What does "visible minority health" mean, and how does it differ from invisible minority health?) Who are visible minorities? I think I know what this means, but the reader should not have to guess. These terms should be explained better in the manuscript, especially because the terms are included in the title and often used in the manuscript.
I'm not an expert in intersectionality, and my work has been mostly focused on explaining differences among countries rather than individuals. The literature review about intersectionality included in this article appears to be up to date and comprehensive. I appreciated that. I'm more familiar with Iris Young's arguments about the five faces of oppression. When I agreed to review this article, I hoped that it would explain the difference between the two arguments, but Young's work is not referenced. I spent some time on the internet, so I now have a better understanding of the difference between her argument and the concept of intersectionality. Maybe I'm just a dinosaur, but I ask that the authors consider including an explanation of the difference between the two conceptions and whether the five faces of oppression conception could also be useful for explaining health inequity.
I'm a quantitative scholar. I was hoping to find some guidance on how to measure intersectionality. Examples and illustrations of alternative approaches would have been useful. As it stands, I wouldn't know how to do it.
Author Response
- I learned from reading this article, but most of what I learned came from the excellent explanation of the concept of intersectionality and how it applies to the analysis of health disparities.
Reply: We sincerely appreciate your positive feedback about our explanation of intersectionality and its application to health disparities analysis. We are pleased that the article provided valuable learning.
- The article does not report the results of new research. It "advocates for thoughtful application of research on visible minority health, urging methodological rigor, contextual awareness, and a focus on actionable interventions." (What does "visible minority health" mean, and how does it differ from invisible minority health?) Who are visible minorities? I think I know what this means, but the reader should not have to guess. These terms should be explained better in the manuscript, especially because the terms are included in the title and often used in the manuscript.
Reply: Thank you for highlighting this critical point. We have provided comprehensive definitions early in the introduction for visible minority and removed the term ‘visible minority health’ considering the creation of ambiguity by this term.
Page 1, lines 39-52:
“Visible minorities” is a term particularly used in Canada to describe non-White, non-Indigenous racialized groups, typically including individuals of Arab, Asian, Black, Hispanic, South Asian, and other diverse ethnic backgrounds, regardless of whether they were born in Canada or immigrated (Employment Equity Act, 1995; Williams et al., 2019). The health inequities in western developed countries are particularly pronounced in these groups which are not isolated phenomena but emerge from complex interactions between multiple social determinants of health including systemic racism, labor market exclusion, language and cultural barriers, and immigration related stressors (Fehrenbacher & Patel, 2020; Harari & Lee, 2021). These determinants differ both in nature and origin from those affecting Indigenous peoples—whose health outcomes are largely shaped by settler colonialism, intergenerational trauma, and land dispossession—and from White populations, who are less likely to encounter racial discrimination or structural exclusion (Parter et al., 2023).
- I'm not an expert in intersectionality, and my work has been mostly focused on explaining differences among countries rather than individuals. The literature review about intersectionality included in this article appears to be up to date and comprehensive. I appreciated that. I'm more familiar with Iris Young's arguments about the five faces of oppression. When I agreed to review this article, I hoped that it would explain the difference between the two arguments, but Young's work is not referenced. I spent some time on the internet, so I now have a better understanding of the difference between her argument and the concept of intersectionality. Maybe I'm just a dinosaur, but I ask that the authors consider including an explanation of the difference between the two conceptions and whether the five faces of oppression conception could also be useful for explaining health inequity.
Reply: We greatly appreciate this suggestion! Young's framework is highly relevant and complementary to intersectionality. Hence we have added this into the theoretical integration section.
Page 10, lines 426-437:
Young's framework of the 'five faces of oppression' offers another valuable theoretical integration with intersectionality for health disparities research (Young, 1988). Young's taxonomy—exploitation, marginalization, powerlessness, cultural imperialism, and violence—provides a structural analysis of the mechanisms through which social inequities are produced and maintained (Young, 2008). When integrated with intersectionality, researchers can identify both who bears compounded burdens of inequity (via intersectionality) and the specific forms that oppression assumes in their lived contexts (via Young’s framework). For example, Ayón et al. (2018) found how the intersecting identities of immigrant Mexican mothers in Arizona created unique vulnerabilities across all five faces of oppression manifested through wage disparities, exclusion from health services, feeling unable to report harassment due to documentation status, shaming for speaking Spanish and traditional childrearing practices, and workplace sexual harassment and more.
I'm a quantitative scholar. I was hoping to find some guidance on how to measure intersectionality. Examples and illustrations of alternative approaches would have been useful. As it stands, I wouldn't know how to do it.
Reply: We understand that it would be great to provide detailed guidance on measuring intersectionality. However, we believe that would require a set of dedicated articles or book chapters for each method. Given our article provides a general overview and discusses different perspectives including both qualitative and quantitative application of this framework and theoretical concepts, we simply indicated different available approaches and indicated how they could be useful or applied. However, we have expanded a bit in this section to clarify and provide more information in respect to your comment.
Page 7-8, lines 312-325:
Mahendran et al. (2022) conducted a comprehensive simulation study comparing six methods for estimating health outcomes across high-dimensional intersections: cross-classification, regression with interaction terms, MAIHDA, classification and regression trees (CART), conditional inference trees (CTree), and random forest. Their findings demonstrate that method choice significantly impacts outcome estimates—in their National Health and Nutrition Examination Survey (NHANES) example, different methods resulted in up to 10 mmHg difference in predicted systolic blood pressure for the same intersectional group, a clinically meaningful difference. For smaller sample sizes, MAIHDA, CTree, and random forest produced more accurate and stable estimates, while CART consistently performed poorly across all sample sizes. Importantly, this evaluation emphasizes that purely data-driven analysis must be balanced with theoretical knowledge about social power structures to ensure that identified intersections remain socially meaningful intervention points rather than merely statistical artifacts
Reviewer 3 Report
Comments and Suggestions for Authors
Thank you for the opportunity to review this manuscript. While the manuscript presents a good review of intersectionality theory, the section on qualitative and quantitative research design considerations is dense and difficult to follow. There is a lot of repetition of points and the authors mostly rely on published research to underscore the importance of incorporating theoretical frameworks into research. The article will hugely benefit from tables or visual aides such as call-out boxes which succinctly summarize the key points they are trying to communicate. Without these, it is difficult to agree with the authors' conclusion that this review "generates crucial insights".
Several strong statements need support. For example, ".... measurement of inequities involve implicit value judgments by researchers in selecting variables, defining categories, and choosing statistical methods, which can bias the findings"
While this assertion is broadly fair, it overlooks the practical constraints faced by researchers working with large secondary datasets. Typically, researchers have little to no influence over what data is collected or how it is processed prior to their access. As a result, even when researchers aim to apply an intersectionality framework in their analyses and interpretations, they may be limited by the structure and content of the available data. It is not fair to conclude that this constitutes as 'value judgements'. Some analytical choices may be made to make the best of what is available. That said, specific examples of when certain variables were neglected in a study when they were clearly available and relevant to outcomes related to inequities would be helpful to drive home this point.
The authors do propose a way to address data limitations through interaction-centered modeling and advocate for the use of MAIHDA as a potential solution. However, the discussion feels incomplete without concrete examples. For instance, are there specific research areas where interaction-centered models have been underutilized? In what ways might MAIHDA enhance the inferences drawn from such analyses? Expanding on these points would meaningfully enrich the discussion and offer genuinely novel insights.
Author Response
- Thank you for the opportunity to review this manuscript. While the manuscript presents a good review of intersectionality theory, the section on qualitative and quantitative research design considerations is dense and difficult to follow. There is a lot of repetition of points and the authors mostly rely on published research to underscore the importance of incorporating theoretical frameworks into research.
Reply: We appreciate your time and effort in reviewing our manuscript and providing constructive feedback. Thank you for identifying these issues with clarity and flow. We have reorganized these sections to reduce repetition and improve readability, while adding more original synthesis beyond citing published research.
- The article will hugely benefit from tables or visual aides such as call-out boxes which succinctly summarize the key points they are trying to communicate. Without these, it is difficult to agree with the authors' conclusion that this review "generates crucial insights".
Reply: We completely agree and have added a table to the revised manuscript to convey key messages of this article.
Page 6 (new Table 1):
Table 1: Key Considerations for Intersectional Research Design
|
Research Phase |
Qualitative Approaches |
Quantitative Approaches |
|
Conceptualization |
• Engage community partners early • Map power relations in research context • Consider historical/political context • Reflexivity about researcher position |
• Move beyond demographic controls • Plan for interaction effects • Consider sample size for subgroups |
|
Data Collection |
• Purposive sampling across intersections • Develop cultural safety protocols • Create spaces for safe expression • Culturally validated instruments |
• Collect disaggregated data • Include contextual variables • Consider proxy measures • Oversample marginalized groups |
|
Analysis |
• Develop intersectional coding schemes • Analyze power in narratives • Look for silences/absences • Use member checking |
• Test interactions systematically • Apply anlaytical method that accommodates intersections • Consider alternative anlaytical method beyond conventions |
|
Interpretation |
• Center marginalized voices • Identify structural barriers • Link to broader systems • Avoid damage-centered narratives |
• Consider practical relevance • Contextualize within power structures • Ensure that the interpretation does not further marginalization |
- Several strong statements need support. For example, ".... measurement of inequities involve implicit value judgments by researchers in selecting variables, defining categories, and choosing statistical methods, which can bias the findings". While this assertion is broadly fair, it overlooks the practical constraints faced by researchers working with large secondary datasets. Typically, researchers have little to no influence over what data is collected or how it is processed prior to their access. As a result, even when researchers aim to apply an intersectionality framework in their analyses and interpretations, they may be limited by the structure and content of the available data. It is not fair to conclude that this constitutes as 'value judgements'
Reply: We acknowledge that this statement needed more nuance and supporting evidence. Further, we agree with the points you made about the constraints of secondary data analysis. We have revised this section to acknowledge these realities while still noting where researcher decisions can make a difference.
Page 7, lines 274-288:
Quantifying intersectionality poses distinct methodological challenges that warrant uptake of innovative analytical approaches. Further, if the researchers are not informed of intersectionality when they design the study and make the analysis plan, their implicit value judgments may be embedded within seemingly objective measurement strategies, potentially obscuring the multidimensional nature of health disparities (Harper et al., 2010). For example, treating “Asian” as a single category conceals critical heterogeneity: Chinese, Filipino, and South Asian subgroups often exhibit markedly different health profiles, yet these differences are lost when data are aggregated into broad racial labels (Sabik, 2021). The selection of reference groups in regression models constitutes another value-laden choice - defaulting to White populations as the normative referent not only centers whiteness in interpretation but also influences the relative magnitude of estimated effects for all other groups (Bowleg, 2012). However, many studies involve working with secondary data where the scope of taking intersectional approaches are often limited due to initial design restrictions.
- Some analytical choices may be made to make the best of what is available. That said, specific examples of when certain variables were neglected in a study when they were clearly available and relevant to outcomes related to inequities would be helpful to drive home this point.
Reply: We have added specific examples for that in response to previous comment.
Page 7, lines 278-287:
For example, treating “Asian” as a single category conceals critical heterogeneity: Chinese, Filipino, and South Asian subgroups often exhibit markedly different health profiles, yet these differences are lost when data are aggregated into broad racial labels (Sabik, 2021). The selection of reference groups in regression models constitutes another value-laden choice - defaulting to White populations as the normative referent not only centers whiteness in interpretation but also influences the relative magnitude of estimated effects for all other groups (Bowleg, 2012). However, many studies involve working with secondary data where the scope of taking intersectional approaches are often limited due to initial design restrictions.
- The authors do propose a way to address data limitations through interaction-centered modeling and advocate for the use of MAIHDA as a potential solution. However, the discussion feels incomplete without concrete examples. For instance, are there specific research areas where interaction-centered models have been underutilized? In what ways might MAIHDA enhance the inferences drawn from such analyses? Expanding on these points would meaningfully enrich the discussion and offer genuinely novel insights.
Reply: Thank you for pushing us to provide more concrete examples and applications. We have added another paragraph drawing from a simulation study by Mahendran et al. 2022.
Page 7, lines 311-323:
Mahendran et al. (2022) conducted a comprehensive simulation study comparing six methods for estimating health outcomes across high-dimensional intersections: cross-classification, regression with interaction terms, MAIHDA, classification and regression trees (CART), conditional inference trees (CTree), and random forest. Their findings demonstrate that method choice significantly impacts outcome estimates—in their National Health and Nutrition Examination Survey (NHANES) example, different methods resulted in up to 10 mmHg difference in predicted systolic blood pressure for the same intersectional group, a clinically meaningful difference. For smaller sample sizes, MAIHDA, CTree, and random forest produced more accurate and stable estimates, while CART consistently performed poorly across all sample sizes. Importantly, this evaluation emphasizes that purely data-driven analysis must be balanced with theoretical knowledge about social power structures to ensure that identified intersections remain socially meaningful intervention points rather than merely statistical artifacts.
Round 2
Reviewer 1 Report
Comments and Suggestions for Authors
Thank you for the opportunity to re-review this article. Overall, I find this to be an improved manuscript that is more comprehensive, as well as clearer in its argument and contributions to the field. I have no further comments for the authors.